# Women's post-abortion contraceptive use: Are predictors the same for immediate and future uptake of contraception? Evidence from Ghana

Esinam Afi Kayi[1]*, Adriana Andrea Ewurabena Biney[2], Naa Dodua Dodoo[2‡], Charlotte Abra Esime Ofori[2‡], Francis Nii-Amoo Dodoo[3]

1 Department of Adult Education and Human Resource Studies, School of Continuing and Distance Education, University of Ghana, Legon, Ghana, 2 Regional Institute for Population Studies, University of Ghana, Legon, Ghana, 3 Department of Sociology, Pennsylvania State University, State College, Pennsylvania, United States of America

☉ These authors contributed equally to this work.
‡ These authors also contributed equally to this work.
* ekayi@ug.edu.gh

**Data Availability Statement:** Data are available from the Measure Demographic and Health Survey program (https://www.dhsprogram.com/data/).

## Abstract

This study seeks to identify the socio-demographic, reproductive, partner-related, and facility-level characteristics associated with women's immediate and subsequent use of post-abortion contraception in Ghana. Secondary data from the 2017 Ghana Maternal Health Survey were utilized in this study. The weighted data comprised 1,880 women who had ever had an abortion within the five years preceding the survey. Binary logistic regression analyses were performed to examine the associations between the predictor and outcome variables. Health provider and women's socio-demographic characteristics were significantly associated with women's use of post-abortion contraception. Health provider's counselling on family planning prior to or after abortion and place of residence were associated with both immediate and subsequent post-abortion uptake of contraception. Among subsequent post-abortion contraceptive users, older women (35–49), women in a union, and women who had used contraception prior to becoming pregnant were strong predictors. Partner-related and reproductive variables did not predict immediate and subsequent use of contraception following abortion. Individual and structural/institutional level characteristics are important in increasing women's acceptance and use of contraception post abortion. Improving and intensifying family planning counselling services at the health facility is critical in increasing contraceptive prevalence among abortion seekers.

## Introduction

Studies indicate that women who have ever terminated a pregnancy are at risk of having more than two abortions in their lifetime [1] and with the risks to women that induced abortions pose, reducing these experiences is important for improving maternal morbidity and mortality

Interested researchers must register on the website to officially request for the specific dataset to have access to the data. Permission to access the data will be granted after the requested information is reviewed. The data is labelled Ghana: Special, 2017 and we used the Individual Raw file (women's individual file) named GHIQ7JFL. It can be found at https://dhsprogram.com/data/dataset/Ghana_Special_2017.cfm?flag=1.

**Funding:** The authors received no specific funding for this work.

**Competing interests:** The authors have declared that no competing interests exist.

globally. The provision of post-abortion contraception immediately following an induced abortion is essential to reducing repeat unwanted pregnancies and induced abortions among women [2]. Evidence from multiple studies indicate that integrating post-abortion services into health care systems increases the contraceptive prevalence rate among women having an abortion [3–8].

In Ghana, contraceptive knowledge is universal, yet less than half of all reproductive aged women (15–49) are using a modern contraceptive method [9]. For instance, according to the 2014 Ghana Demographic and Health Survey (GDHS), the proportion of both married and unmarried women using a modern method of contraception differs. About 18.2% of all women were currently using a modern method of contraception [9].

The 2007 and 2017 Ghana Maternal Health Surveys (GMHS) provide relevant information on maternal, and child health indicators as well as the reproductive histories of women five years preceding the survey [10,11]. From both surveys, the proportion of reproductive aged women (15–49) who had an induced abortion in the five years preceding the survey was relatively unchanged between 5% in 2007 and 7% in 2017. The 2007 GMHS report indicates that, out of the 28% of women who used a contraceptive method prior to the pregnancy resulting in an abortion, 20% reported using a modern type of contraceptive. On the other hand, of the women who had an induced abortion during the five-year period preceding the 2017 GMHS, only 20% used some form of contraception before becoming pregnant and undergoing a subsequent abortion [11]. Additionally, both surveys report the contraceptive prevalence rate among women who had an induced abortion. In 2017, 36% of women received a contraceptive method after induced abortion by a health personnel while this figure was lower (5%) in 2007 [10,11]. The findings from these nationally representative health surveys suggest that multiple factors account for the low use of modern contraceptive methods before and after women's induced abortion experiences. It is possible that among women who underwent an induced abortion, few may have encountered health personnel at the point of abortion care where family planning services were offered; hence, the increased uptake of contraception post abortion during the ten-year period. For women who have ever had unplanned pregnancies that resulted in abortion, the imperative to use contraception to prevent another unplanned pregnancy may be greater.

To the best of our knowledge, no studies to date have used nationally representative data to investigate the range of factors that would predict women's use of contraception following an abortion in Ghana. Thus, it is not clear whether demographic factors alone predict family planning use after abortion among women in Ghana. There is also sparse literature from the Ghanaian context that indicate which reproductive factors, partner-related and health facility-related characteristics influence such contraceptive practices among women who have had an abortion in their lifetime. Previous studies that examine post-abortion contraception tend to use qualitative methods, eliciting data from abortion seekers, healthcare providers and community members on experiences and perceptions of post-abortion contraceptive uptake [12] or use health facility data focusing on facility related variables [8,13]. These studies we believe, are insufficient in unravelling the multiple factors that may be critical in improving women's uptake of contraception following abortion.

Using the 2017 GMHS, we investigate the factors that account for women's immediate and subsequent or future use of contraception following abortion, considering socio-demographic, reproductive, partner-related, and facility-level factors. Several studies have investigated factors associated with contraceptive use among women in reproductive ages [14–18] but, abortion seekers have unique characteristics that put them at risk of experiencing multiple abortions. Abortion seekers in Ghana are younger, reside in urban areas, have a higher socio-economic status, and have no child or few children [19]. On the other hand, women who

undergo unsafe abortions are typically rural residents, less educated, and have a lower socio-economic status [19,20].

This study provides a novel means to examine the determinants associated with contraceptive uptake post-abortion in order to highlight the multiple factors needed to increase opportunities for post-abortion women to have access to family planning services with the ultimate goal of reducing the risk of repeat abortions. In a context where unmet need for contraception is high [21,22] and abortions occur repeatedly, especially, among young women who are more likely to induce abortions unsafely [19], it is important to investigate the multiple level characteristics associated with the adoption of post-abortion contraception among abortion seekers.

## Socio-ecological model explaining contraceptive behavior among abortion seekers

Previous studies on the determinants of contraceptive use after abortion reveal varying results due to contextual differences, study designs and outcomes of interests [5,23–26]. Research also shows that contraceptive uptake is highest when immediately offered on-site, or within less than six months or up to a year [23,26–29]. For instance, Adelman, Free and Smith [29] examined the predictors of post-abortion contraception at four and 12 months among Cambodian women. They found that previous contraceptive use, intention to use contraception, and number of living children were strongly associated with post-abortion contraception use. Occupation was significantly associated at four months whilst abortion method predicted post-abortion contraception at 12 months.

In an intervention study conducted by Johnson and colleagues [23], their results showed that women who received post-abortion family planning services during an intervention programme had significantly lower unplanned pregnancies and few repeat abortions during a 12 month follow-up period compared to post-abortion women who were not included in the intervention programme. In addition, Delvaux and colleagues [30] found that post-abortion contraceptors were more likely to be older, married, have at least one child and use contraception before the abortion.

Similarly, other studies have reported greater acceptance and/or use of modern contraceptives among women who received PAFP counselling [31] and services if they attended health centers or maternity homes compared to hospitals [32,33], and if women attended urban hospitals compared to rural hospitals or attended Protestant hospitals compared with Catholic hospitals [34]. Women were also more likely to use modern contraception after abortion if the family planning counselling and services was provided by gynaecological ward staff compared to other models of provision [3].

The socio-ecological model can best explain factors associated with contraceptive uptake among women with abortion experiences. Studies have explored factors associated with contraceptive use at the individual (intrapersonal), partner (interpersonal) and institutional or structural levels [35]. Age, education, religion, marital status, ethnicity, number of living children, number of abortions and personal motivation for abortion are individual-centred factors that can determine use among women. While some studies have found that older women are more likely than younger women to use contraception, especially long-acting methods, because of prior experience and the desire to limit childbearing [13], others have found that younger women are more likely to adopt contraception post-abortion [8,30,36]. Also, women with some level of education may have a greater likelihood of contraceptive adoption than uneducated women while women belonging to all other religious affiliations other than Catholics and all other ethnic group other than Mole-Dagbanis are predisposed to use a contraceptive method [12,13].

Important interpersonal factors have included those associated with the male-partner. The few studies on this have discussed that financial, communicative, and emotional support are predictors of contraception uptake immediately post-abortion [37]. Men's participation and support during the abortion process may be linked to women's acceptance and use of contraception post abortion [12,37,38]. In the literature, institutional level factors are characterized by facility-level factors and these are also key to understanding contraceptive use. The abortion procedure, facility type and type of provider may determine use through these mechanisms [13]. Structural level factors may also determine access to facilities for contraceptive use [28].

The two aims of this study are to examine the key factors associated with immediate post-abortion contraceptive use, as well as those associated with subsequent or future use, suggesting sustained use. We argue that abortion seekers in Ghana may use post-abortion contraception when partner support and health provider or facility-level factors are available.

## Materials and methods

### Study design and setting

The 2017 Ghana Maternal Health Survey dataset is a publicly accessible document which was obtained from www.DHSprogram.com [11]. The survey was jointly implemented by the Ghana Statistical Service (GSS) and the Ghana Health Service (GHS) with technical assistance from the ICF (originally Inner City Fund) through the DHS program. Participants in the survey provided written informed consent and participation in the survey was voluntary. The dataset was completely anonymized, de-identified and aggregated before we were provided with access to it for analysis. The identities of participants were therefore, not linked to the data. The 2017 GMHS is the second nationally representative household two-stage cluster survey which comprises comprehensive information on maternal and reproductive health issues, maternal mortality, and specific causes of death among women in Ghana [11]. The survey protocols and biomarkers were reviewed and approved by the ICF Institutional Review Board. The data that were collected comprised events within five years preceding the survey. The women's individual dataset used for this study was limited to a weighted sample of 1,880 women (both married and unmarried) who had ever had an abortion within the five years preceding the survey.

### Measures

The primary outcome variables were immediate and subsequent or later post-abortion family planning or contraceptive use. In the dataset, immediate post-abortion contraceptive use was captured by two questions. The first question asked respondents, 'After this abortion, did the doctor or health worker give you a method of contraception, prescribed a method of contraception, or refer you to a family planning clinic?' with responses as either 'Yes' or 'No'. a subsequent question for those responding in the affirmative was "Did they give you the method of contraception, give you a prescription, or give you a referral?" and the corresponding options were 'Gave method', 'Prescribed a method', 'Gave a referral' or 'Don't know'. Respondents who chose the option 'gave method' were categorised as immediate users and were coded as 'Yes' while all the other options were categorised as 'No'. Given the data limitation of questions on the immediacy of actual contraceptive use post-abortion, we assume that those 'given a method' suggests acceptance of and actual uptake of a contraceptive method after the abortion. Subsequent post-abortion uptake of contraception was measured by women's current use of contraception at the time of the survey. In this study, modern contraceptive users should be using implants, intrauterine device (IUD), oral contraceptive pills, injectables, male and female condoms. All others were coded as 'No'.

The predictor variables in the study included socio-demographic, reproductive, and abortion-related service characteristics, partner factors, and health facility indicators. The socio-demographic characteristics examined include age, highest educational status, religious affiliation, ethnicity, marital status, and place of residence. The reproductive factors were number of abortions (1, 2, 3+) and contraceptive use at index pregnancy (No/Yes). Abortion-related service delivery characteristics consisted of pre-or post-abortion health provider counselling and services (No/Yes) and abortion method (Safe/Unsafe). Respondents' abortion method was categorised as safe or unsafe based on the WHO classification (place abortion was performed, category of health provider who performed the abortion and equipment used). Partner-related factors were measured based on three variables: attitude towards the abortion (Favourable/Opposed), provision of financial support for the abortion (Yes/No), and partner-related reasons for the abortion (Health-related, Partner-related and Other-related).

## Data analysis

The data were analysed in three stages. First, we employed univariate techniques to explore frequencies of the categorical variables. Second, for the bivariate analyses, descriptive statistics were reported for all categorical and continuous predictor variables with their differences assessed by selected characteristics using cross-tabulations and chi-square tests. Third, multivariate analyses were conducted using two binary logistic regression models to ascertain the odds of all the factors significantly associated with immediate and subsequent post-abortion contraceptive use. All analyses were performed using STATA version 13.

There were few limitations which pertained to restrictions with the dataset in measuring some variables. First, we were unable to measure immediate post-abortion contraceptive use due to the limited questions on actual contraceptive use immediately post-abortion; hence, the assumption that 'given method' is actual immediate contraceptive use. Second, another limitation was our inability to measure some individual characteristics such as occupation and number of surviving children. In addition, the type of health facility—primary, secondary, or tertiary—where women obtained induced abortion could not be ascertained more specifically for and considered in the analysis since it was not captured in the data.

## Results

Among the sample of women who had ever had an abortion in the five years preceding the survey, the majority were more than 25 years old. The mean age of respondents was 27 years. Most respondents (89%) had received formal education with only 11% having never had any formal education. The study population was predominantly Christians and exactly half of the sample belonged to the Pentecostal/Charismatic faith. Concerning ethnicity, the majority were Akans (55%) and the least represented ethnic group was Ga (7%). Less than one-third of women had been using a contraceptive method prior to the index pregnancy that ended in the abortion. Only 30% of the sample practiced a safe abortion as their last pregnancy termination method. About 6.9% of women used contraception immediately after the abortion while women who delayed uptake until later were 48%. The percentage of women using specific contraceptive methods post-abortion subsequently varied: 49% were using short-acting contraceptives; 21% were using long-acting reversible contraception, and 19% preferred traditional methods. The remaining were currently using emergency contraceptive pills as their main method of pregnancy prevention (specific methods not shown in table). The key characteristics of the sample are presented in Table 1.

Results from the bivariate analysis (Table 2) showed seven factors associated with subsequent uptake of contraception post-abortion. A little over half the proportion of women aged

**Table 1. Participants' characteristics (N = 1880).**

| Variables | Categories | N | Percent (%) |
|---|---|---|---|
| *Socio-demographic factors* | | | |
| Age | 15–24 | 694 | 36.9 |
| | 25–34 | 867 | 46.1 |
| | 35–49 | 319 | 17.0 |
| Highest educational level | No education | 161 | 8.6 |
| | Primary | 298 | 15.9 |
| | Junior High School | 874 | 46.5 |
| | Senior High School | 410 | 21.8 |
| | Tertiary | 137 | 7.3 |
| Religious affiliation | Catholic | 164 | 8.7 |
| | Orthodox | 241 | 12.8 |
| | Charismatic/Pentecostal | 1043 | 55.5 |
| | Islam | 266 | 14.1 |
| | Other Christian | 128 | 6.8 |
| | No religion | 38 | 2.1 |
| Ethnicity | Akan | 1133 | 60.3 |
| | Ga | 171 | 9.1 |
| | Ewe | 286 | 15.2 |
| | Mole Dagbani | 142 | 7.6 |
| | Other ethnic groups | 148 | 7.9 |
| Marital status | Not in union | 760 | 40.5 |
| | In union | 1120 | 59.6 |
| Place of residence | Urban | 1229 | 65.4 |
| | Rural | 651 | 34.6 |
| *Reproductive factors* | | | |
| Number of abortions | 1 | 1269 | 67.5 |
| | 2 | 455 | 24.2 |
| | 3+ | 156 | 8.3 |
| Contraceptive use at index pregnancy | No | 1528 | 81.3 |
| | Yes | 352 | 18.7 |
| *Abortion-related service delivery factors* | | | |
| Type of abortion | Safe | 657 | 34.9 |
| | Unsafe | 1223 | 65.1 |
| Health provider counselling on family planning before/ after abortion | Yes | 596 | 31.7 |
| | No | 1284 | 68.3 |
| *Partner-related factors* | | | |
| Partner attitude towards abortion | Opposed | 887 | 47.2 |
| | Favourable | 993 | 52.8 |
| Partner paid for abortion | Yes | 844 | 44.9 |
| | No | 1036 | 55.1 |
| Main reason for abortion | Health-related | 143 | 7.6 |
| | Partner-related | 278 | 14.8 |
| | Other reasons | 1459 | 77.6 |
| **Outcome variables** | | | |
| Immediate post-abortion contraceptive use | No | 1750 | 93.1 |
| | Yes | 130 | 6.9 |
| Subsequent post-abortion contraceptive use | No | 989 | 52.6 |

(*Continued*)

**Table 1.** (Continued)

| Variables | Categories | N | Percent (%) |
|---|---|---|---|
| | Yes | 891 | 47.4 |
| Total | | 1880 | 100.0 |

15–24 years reported that they used contraception subsequently post-abortion while less than half of women in older age groups reported the same. More than half (55%) of women in rural areas and 44% of urban women adopted contraception following abortion subsequently. Also, exactly half the proportion of women in union reported using contraception subsequently after the abortion (p-value = 0.04), while over four out of ten women currently not in union used contraception. Contraceptive use at index pregnancy was the only reproductive history factor associated with women's subsequent use of contraception. About 6 in 10 respondents who used contraception at index pregnancy also reported that they used contraceptives later post-abortion (p-value = <0.001).

All the partner-related characteristics were significantly associated with women's uptake of contraception after the abortion subsequently. For instance, results indicate that 45% of women who reported that their partners had favourable attitudes towards the abortion used contraception subsequently following the abortion (p-value = 0.02). Further, more than half (52%) the proportion of women who indicated that their partners paid for the abortion used contraception subsequently after the abortion. Regarding the main reason for the abortion, about 42% of women who stated that some partner related reasons were the main reasons for the abortion used contraception subsequently post-abortion. The smallest proportion of respondents using contraception were in the 'health-related' reason category (36%).

Health provider counselling on family planning prior to or before the abortion, and place of residence were the only variables significantly associated with women's uptake of immediate post-abortion contraception. Ten percent of women in rural areas reported use of contraception immediately while five percent of rural respondents reported the same. In addition, close to one in five women who received counselling on family planning before/after abortion from a health provider reported using contraception immediately after the abortion, while only one percent of women with no counselling used contraception immediately post-abortion.

Table 3 shows the results of the multivariate regression analyses for immediate and subsequent post-abortion family planning use. Among immediate post-abortion contraceptive users, only place of residence and counselling on a family planning method by a health provider before or after abortion was significant (Model 1). Women residing in rural areas were more likely to use contraception immediately following an abortion compared to women in urban areas (OR = 1.79; 95% CI = 1.088–4.249). Women were less likely to use family planning immediately after the pregnancy termination if they did not receive family planning counselling from a health professional before or after the abortion compared to women who were counselled by a health professional prior to or post abortion (OR = 0.04; 95% CI = 0.021–0.079).

Results from Model 2 in Table 3 show that the determinants significantly associated with subsequent post-abortion contraceptive uptake include age, marital status, place of residence, contraceptive use at index pregnancy, and receipt of pre or post FP counselling by a health provider. The results indicate that compared to young women (aged 15–24 years), older women (between 35–49 years) were less likely to initiate post-abortion contraception in future (OR = 0.38; 95% CI = 0.253–0.572). Also, compared to single women, women in a union were 1.41 times as likely to adopt a post abortion contraceptive method subsequently (OR = 1.41; 95% CI = 1.086–1.834). Prior use of contraception before the abortion (contraceptive use at index pregnancy) predicted

**Table 2. Distribution of predictor variables by immediate and subsequent post-abortion family planning (PAFP) uptake.**

| Variables | Categories | Immediate PAFP | | | Subsequent PAFP | | |
|---|---|---|---|---|---|---|---|
| **Socio-demographic factors** | | Yes | $\chi^2$ | p-value | Yes | $\chi^2$ | p-value |
| Age | 15–24 | .07 | 1.339 | .57 | .53 | 32.719 | .000 |
| | 25–34 | .06 | | | .49 | | |
| | 35–49 | .07 | | | .31 | | |
| Highest educational level | No education | .06 | 7.336 | .13 | .46 | .722 | .97 |
| | Primary | .11 | | | .46 | | |
| | Junior High School | .07 | | | .47 | | |
| | Senior High School | .06 | | | .49 | | |
| | Tertiary | .03 | | | .49 | | |
| Religious affiliation | Catholic | .05 | 2.515 | .69 | .54 | 4.006 | .69 |
| | Orthodox | .07 | | | .51 | | |
| | Charismatic | .08 | | | .46 | | |
| | Islam | .06 | | | .47 | | |
| | Other Christian | .06 | | | .48 | | |
| | No religion | .03 | | | .46 | | |
| Ethnicity | Akan | .07 | 2.787 | .63 | .46 | 7.778 | .21 |
| | Ga | .08 | | | .55 | | |
| | Ewe | .05 | | | .42 | | |
| | Mole Dagbani | .08 | | | .55 | | |
| | Other ethnic groups | .09 | | | .49 | | |
| Residence | Urban | .05 | 14.398 | .00 | .44 | 10.288 | .007 |
| | Rural | .10 | | | .53 | | |
| Marital status | Not in union | .07 | .227 | .67 | .44 | 5.927 | .04 |
| | In union | .07 | | | .50 | | |
| **Reproductive factors** | | | | | | | |
| Number of abortions | 1 | .07 | 1.839 | .46 | .49 | 3.002 | .29 |
| | 2 | .06 | | | .45 | | |
| | 3+ | .09 | | | .43 | | |
| Contraceptive use at index pregnancy | No | .06 | .806 | .40 | .45 | 17.108 | .000 |
| | Yes | .08 | | | .59 | | |
| **Abortion service delivery factors** | | | | | | | |
| Type of abortion | Safe abortion | .08 | 2.392 | .14 | .43 | 5.214 | .06 |
| | Unsafe abortion | .06 | | | .49 | | |
| Health provider counselling on FP before/ after abortion | Yes | .19 | 161.05 | .000 | .51 | 4.208 | 0.10 |
| | No | .01 | | | .45 | | |
| **Partner-related factors** | | | | | | | |
| Partner attitude towards abortion | Opposed | .07 | .05 | .83 | .44 | 6.721 | .02 |
| | Favourable | .07 | | | .51 | | |
| Partner paid for abortion | Yes | .07 | .022 | .88 | .52 | 9.844 | .03 |
| | No | .07 | | | .44 | | |
| Main reason for abortion | Health-related | .06 | .355 | .82 | .36 | 9.716 | .04 |
| | Partner-related | .08 | | | .42 | | |
| | Other reasons | .07 | | | .49 | | |

subsequent use following abortion (OR = 1.74; p<0.05). Like immediate post abortion contraceptors, if women were not offered FP counselling pre/post abortion by a health professional, they were less likely to use FP subsequently after the abortion (OR = 0.73; p<0.05).

**Table 3. Associations between predictor variables and immediate and subsequent post-abortion family planning (PAFP) uptake.**

| Variables | Categories | Model 1 Immediate PAFP | | Model 2 Subsequent PAFP | |
|---|---|---|---|---|---|
| **Socio-demographic factors** | | | | | |
| | | OR | 95 CI% | OR | 95 CI% |
| Age | 15–24 [RC] | 1.00 | | 1.00 | |
| | 25–34 | 0.60 | .322–0.121 | 0.84 | .645–1.092 |
| | 35–49 | 0.65 | .309–1.353 | 0.38** | .253-.572 |
| Highest educational level | No education [RC] | 1.00 | | 1.00 | |
| | Primary | 2.03 | .901–4.581 | 0.96 | .540–1.689 |
| | Junior High School | 1.03 | .470–2.242 | 0.91 | .524–1.586 |
| | Senior High School | 0.73 | .309–1.709 | 0.91 | .517–1.601 |
| | Tertiary | 0.51 | .128–2.066 | 1.04 | .513–2.126 |
| Religious affiliation | Catholic [RC] | 1.00 | | 1.00 | |
| | Orthodox | 1.54 | .530–4.453 | 0.84 | .512–1.367 |
| | Charismatic | 1.53 | .648–3.638 | 0.70 | .454–1.081 |
| | Islam | 1.13 | .431–2.945 | 0.77 | .463–1.278 |
| | Other Christian | 1.00 | .334–2.970 | 0.74 | .361–1.413 |
| | No religion | 0.43 | .038–4.902 | 0.64 | .241–1.696 |
| Ethnicity | Akan [RC] | 1.00 | | 1.00 | |
| | Ga | 1.00 | .403–2.464 | 1.50* | .956–2.349 |
| | Ewe | 0.71 | .323–1.564 | 0.94 | .653–1354 |
| | Mole Dagbani | 1.43 | .672–3.043 | 1.27 | .746–2.178 |
| | Other | 1.43 | .482–4.249 | 1.02 | .604–1.736 |
| Residence | Urban [RC] | 1.00 | | 1.00 | |
| | Rural | 1.79** | 1.088–2.954 | 1.35** | 1.021–1.784 |
| Marital status | Not in union [RC] | 1.00 | | 1.00 | |
| | In union | 1.00 | .601–1.677 | 1.41** | 1.086–1.834 |
| **Reproductive factors** | | | | | |
| Number of abortions | | 1.10 | .859–1.419 | 1.00 | .872–1.135 |
| Contraceptive use at index pregnancy | No [RC] | 1.00 | | 1.00 | |
| | Yes | 1.17 | .632–2.169 | 1.74** | 1.215–2.507 |
| **Abortion service delivery factors** | | | | | |
| Type of abortion | Safe [RC] | 1.00 | | 1.00 | |
| | Unsafe | 1.07 | .661–1.726 | 1.32* | .986–1.762 |
| Health provider counselling on FP before/ after abortion | Yes [RC] | 1.00 | | 1.00 | |
| | No | 0.04** | .021-.079 | 0.73** | .542-.990 |
| **Partner-related factors** | | | | | |
| Partner attitude towards abortion | Opposed [RC] | 1.00 | | 1.00 | |
| | Favourable | 1.08 | .626–1.851 | 1.25 | .950–1.657 |
| Partner paid for abortion | Yes [RC] | 1.00 | | 1.00 | |
| | No | 0.93 | .540–1.605 | 0.84 | .605–1.177 |
| Main reason for abortion | Health-related [RC] | 1.00 | | 1.00 | |
| | Partner-related | 1.56 | .548–4.442 | 1.22 | .644–2.306 |
| | Other reasons | 1.28 | .572–2.870 | 1.49 | .889–2.486 |

**p<0.05

*p<0.1.

## Discussion

In this study, we examined the factors associated with immediate and subsequent or future use of modern contraception following induced abortion among a retrospective cohort of women. The study findings are discussed below on the levels within the socio-ecological framework which guided the study.

### Individual level factors of post-abortion contraceptive use

At the individual/intrapersonal level, age, marital status, place of residence, and contraceptive use before pregnancy were associated with subsequent/future post-abortion contraceptive use. Age as a determinant of post-abortion contraception use has been found in other studies [8,30,36] and these findings are consistent with our study which showed that older women (>35 years) were less likely to use contraception subsequently after abortion compared to adolescents and young women. Attitudinal resistance and prior experience with contraceptive use may account for older women's unwillingness to initiate contraception following an abortion. Fertility intentions and reproductive goals of women differ. Similarly, continuation rates of contraception uptake after abortion may vary for younger and older women. Young women (including adolescents) may be more willing to accept family planning to prevent future unplanned pregnancies because they may have educational aspirations and other life achievement goals to pursue. It is also likely that young women are inexperienced with modern contraception and perhaps more susceptible to 'subtle directives' from health providers to accept post-abortion family planning counselling services compared to older women.

The findings also show that, compared to women who were not in a union, women in a union were more likely to use contraception at any time following termination of pregnancy. This finding is somewhat unexpected although there is evidence to demonstrate that married women initiate post-abortion contraception with their partners' approval [39]. It is plausible that married women have reached their fertility goals and aspirations, therefore, they prefer using contraception to regulate or stop childbearing [40]. On the other hand, unmarried women may have fears or anxieties about adopting contraception if they believe that using contraceptives will make them infertile before they get married. Support and involvement of male partners at the time of the abortion could also influence the use of contraception after abortion [37].

From the results of the study, place of residence is an important factor in women's use of modern contraception immediately and subsequently following induced abortion. We found that compared to women residing in urban areas, women living in rural settings were more likely to use modern family planning methods post-abortion. Similar findings have been reported in other contexts among women in their reproductive ages [41,42]. In Ghana for instance, the 2007 GMHS evidences slight differences in modern family planning use by women living in urban and rural areas. Aviisah and colleagues [43] used the 2003, 2008 and 2014 demographic and health surveys to examine patterns in the use of modern contraceptive methods among married women. They found an increase in modern methods of contraceptive use among married women in rural areas compared to urban dwellers. Women living in rural settings in Ghana have greater avenues and opportunities to access, use, and choose from a variety of modern family planning methods due to the implementation of the Community-based Health Planning and Services (CHPS) concept at the district levels which expands and integrates the provision of child and maternal care with reproductive health services [44].

Prior contraceptive use before pregnancy termination is necessary to sustaining efforts aimed at avoiding future unplanned pregnancies. We found that women were more likely to use contraception not immediately but, subsequent after an induced abortion, if they had used

contraceptives prior to becoming pregnant. Findings echo results from other settings indicating that contraceptive history predicts post abortion contraceptive use [26,45,46]. These findings can be explained considering women's desire and intention to reduce their fertility or stop having children as well as discontinuing future unplanned pregnancies. By implication, the previous contraceptive practices of women present an opportunity for health providers to counsel women on the efficacy and benefits of effective long-acting contraceptive methods at the time of abortion.

### Interpersonal factors associated with post-abortion contraceptive use

According to the socio-ecological model, an individual's decision is shaped not only by their own characteristics, but by others including significant others. Kayi [37] and Rominski et al [12] demonstrated that attributes of male partners shaped women's reproductive decisions. Our results, however, show that partner-related characteristics such as male partner attitude, financial support of male partner and partner-related reasons for the abortion were not associated with immediate and subsequent post-abortion contraception use. These findings provide insights for discussions on autonomy in contraceptive decision-making among post-abortion women who have varying degrees of vulnerability [20]. Though evidence exists to demonstrate male dominance in women's contraceptive decisions, male power appears to be overshadowed during abortion care.

### Structural/Institutional level- health system factors

The findings show that structural level factors specifically, the provision of family planning counselling by health providers before or after abortion is strongly associated with immediate and future use of post-abortion contraception. Our study found that, women were less likely to use contraception following abortion if family planning counselling was not provided by a health provider. This finding suggests that, abortion method, type of facility, and location (or access) are not as important as the role played by health professionals in the delivery of abortion care and family planning services. Health care providers form an integral part of the healthcare system particularly where maternal and reproductive health services are concerned. To a large extent, confidence in the medical expertise of health professionals coupled with a change in provider attitudes on abortion, training and provider skills [47] may have served as strong motivating factors to encourage women's adoption and uptake of a contraceptive method post-abortion. The results are consistent with existing studies [3,13,32,33,48].

## Conclusions

This study contributes to the growing body of post-abortion research, specifically, on the determinants of contraceptive uptake in the immediate post-abortion period and subsequently within a five-year period among Ghanaian women. We find that individual (intrapersonal) and structural/institutional level characteristics are significant in predicting women's use of contraception following abortion either immediately or in future. Except for woman's age, marital status, and place of residence, all the other individual level factors are not associated with post-abortion FP use. Structural/institutional level factors have the potential to influence women's intention and continued use of FP, but health professionals' provision of pre/post abortion FP counselling is a sine qua non. Findings from this study are relevant and informative in building the evidence base of institutional relevant factors influencing post-abortion care in Ghana. Policy makers involved in comprehensive abortion care delivery need to target young women when designing programmes to increase the contraceptive prevalence rate among post-abortion women. Expanding access to and availability of post-abortion FP

counselling and services, equipping health providers on abortion care, in addition to the availability of a variety of modern contraceptives at the time of abortion care are key to increasing the contraceptive prevalence rate among post-abortion women.

## Acknowledgments

The authors are grateful to the writing group team for their intellectual contributions towards this paper.

## Author Contributions

**Conceptualization:** Esinam Afi Kayi, Naa Dodua Dodoo.

**Formal analysis:** Esinam Afi Kayi, Adriana Andrea Ewurabena Biney, Charlotte Abra Esime Ofori.

**Methodology:** Esinam Afi Kayi, Adriana Andrea Ewurabena Biney, Charlotte Abra Esime Ofori.

**Project administration:** Naa Dodua Dodoo.

**Resources:** Francis Nii-Amoo Dodoo.

**Supervision:** Adriana Andrea Ewurabena Biney, Naa Dodua Dodoo, Francis Nii-Amoo Dodoo.

**Writing – original draft:** Esinam Afi Kayi.

**Writing – review & editing:** Esinam Afi Kayi, Adriana Andrea Ewurabena Biney, Naa Dodua Dodoo, Charlotte Abra Esime Ofori, Francis Nii-Amoo Dodoo.

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
