## [Decision Letter · Decision Letter 0]

21 Oct 2020

PONE-D-20-22759

Women’s post-abortion contraceptive use: are predictors the same for immediate and future uptake of contraception? Evidence from Ghana.

PLOS ONE

Dear Dr. Esinam Afi Kayi,

Thank you for submitting your manuscript to PLOS ONE. After careful consideration, we feel that it has merit but does not fully meet PLOS ONE’s publication criteria as it currently stands. Therefore, we invite you to submit a revised version of the manuscript that addresses the points raised during the review process.

Kindly revise your manuscript taking into cognisance the comments of Reviewer 2. 

We look forward to receiving your revised manuscript.

Kind regards,

Eugene Kofuor Maafo Darteh, Ph.D.

Academic Editor

PLOS ONE

Journal Requirements:

2. Please change your reference to "p=0.000" to "p<0.001" or as similarly appropriate, as p values cannot equal zero.

3. In ethics statement in the manuscript and in the online submission form, please provide additional information about the database used in your retrospective study. Specifically, please ensure that you have discussed whether all data were fully anonymized before you accessed them and/or whether the IRB or ethics committee waived the requirement for informed consent. If patients provided informed written consent to have their data used in research, please include this information.

Reviewers' comments:

Reviewer's Responses to Questions

**Comments to the Author**

1. Is the manuscript technically sound, and do the data support the conclusions?

Reviewer #1: Yes

Reviewer #2: Yes

2. Has the statistical analysis been performed appropriately and rigorously? 

Reviewer #1: Yes

Reviewer #2: Yes

3. Have the authors made all data underlying the findings in their manuscript fully available?

Reviewer #1: Yes

Reviewer #2: Yes

4. Is the manuscript presented in an intelligible fashion and written in standard English?

Reviewer #1: Yes

Reviewer #2: Yes

5. Review Comments to the Author

Reviewer #1: The paper was generally well written

The background information was well researched and unambiguous. The socio-ecological model used to underpin the study was appropriate and commendable.The objectives set were to examine key factors associated with immediate post-abortion contraceptive use and those associated with subsequent or future use. These are clear and unambiguous

Secondary data was used in this study. The methods used in the primary data collection were summarised in the paper. A multi-stage sampling technique was employed. The logistic regression analysis employed was appropriate and rigorous.

Findings generally reflected objectives set, however, the finding that about 9% of women used contraception immediately after the abortion was wrongly reported. It is 6.9% and not 9% (see table 1).

The conclusions drawn and recommendations made generally reflected the objectives set and gaps respectively observed.

Reviewer #2: 1. In the abstract, the recommendation is not specific. Again, the study did not focus on access to modern contraceptives and so the related recommendation is misplaced. Similarly, the study did not look into access to post-abortion contraceptives, hence, a wild recommendation.

2. At line 72 and 73, the statement that less than one-quarter of women in the reproductive ages (15-49) use any method of contraception is misleading: The key variable is women currently married. Again, figure is not less than one-quarter. More so, it is not ‘any method’ of contraceptive. The authors could rather include evidence of contraceptive use and induced abortion among such cohort.

3. Data on the contraceptive 78 prevalence rate (CPR) of women who have ever had an abortion is lacking’ – this statement made on line 77 was not supported with any evidence. Apart from the GDHS (1988-2014), the GMHS (2007 and 2017) data are available.

4. The justification for the study is presented at line 77-88. The authors admit they did not have any nationally representative study on the subject. What about other jurisdictions – especially Africa, and other regions - Asia, Europe, Americas? Again, they reported that there are qualitative studies; they need to explore further to raise justifiable arguments.

5. There is no commentary on the findings based on the GMHS 2007 and 2017 at the introduction section. This makes the section scanty on relevant issues.

6. In discussing the subject with the socio-ecological model (Lines 132-160), the authors indicated three levels of interests: individual (intrapersonal), partner (interpersonal) and institutional or structural levels. The authors must therefore put the discussion under each of these interests to interpret it within the context of the model.

7. At Lines 154-156, the authors indicate that the study is an explanatory study. This is questionable because there are not additional qualitative information to complement the predictor information.

8. Information at Lines 158-160 defeat the earlier position of the author with regard to the justification of the study at Lines 77-88.

9. There are no references under the Material and Methods section; not even the source of the data sets. This is unethical or suggests plagiarism.

10. The socio-ecological model should be discussed with the findings. This was not done at the discussion level.

6. PLOS authors have the option to publish the peer review history of their article (what does this mean?). If published, this will include your full peer review and any attached files.

Reviewer #1: No

Reviewer #2: No

---

## [Author Response · Author response to Decision Letter 0]

25 Jan 2021

Reviewer #1: 

The paper was generally well written

The background information was well researched and unambiguous. The socio-ecological model used to underpin the study was appropriate and commendable. The objectives set were to examine key factors associated with immediate post-abortion contraceptive use and those associated with subsequent or future use. These are clear and unambiguous

Response: We are grateful to the reviewer for these positive comments

Secondary data was used in this study. The methods used in the primary data collection were summarised in the paper. A multi-stage sampling technique was employed. The logistic regression analysis employed was appropriate and rigorous.

Findings generally reflected objectives set, however, the finding that about 9% of women used contraception immediately after the abortion was wrongly reported. It is 6.9% and not 9% (see table 1).

The conclusions drawn and recommendations made generally reflected the objectives set and gaps respectively observed.

Response: We are grateful for the reviewer catching this error. This has been rectified in table 1.

Reviewer #2: 

1. In the abstract, the recommendation is not specific. Again, the study did not focus on access to modern contraceptives and so the related recommendation is misplaced. Similarly, the study did not look into access to post-abortion contraceptives, hence, a wild recommendation.

Response: The recommendation in the abstract has been rephrased to suit the findings from the study. We are grateful for this observation. Please see lines 59-62.

2. At line 72 and 73, the statement that less than one-quarter of women in the reproductive ages (15-49) use any method of contraception is misleading: The key variable is women currently married. Again, figure is not less than one-quarter. More so, it is not ‘any method’ of contraceptive. The authors could rather include evidence of contraceptive use and induced abortion among such cohort.

Response: The misleading information has been corrected based on evidence from the relevant references. The study consists of all reproductive aged (15-49) women (both married and unmarried) with an induced abortion in the 5 years preceding the survey. The suggestion for evidence of contraceptive use and induced abortion among the subgroup of women has been incorporated into the manuscript at lines 74-77. 

3. Data on the contraceptive 78 prevalence rate (CPR) of women who have ever had an abortion is lacking’ – this statement made on line 77 was not supported with any evidence. Apart from the GDHS (1988-2014), the GMHS (2007 and 2017) data are available.

Response: We are grateful to the reviewer for this critical observation. The statement that the contraceptive prevalence rate (CPR) of women who have ever had an abortion is lacking has been removed. Data from the 2007 and 2017 Ghana Maternal and Health Surveys have been incorporated into the introduction sections of the manuscript. We agree that the data are available. 

4. The justification for the study is presented at line 77-88. The authors admit they did not have any nationally representative study on the subject. What about other jurisdictions – especially Africa, and other regions - Asia, Europe, Americas? Again, they reported that there are qualitative studies; they need to explore further to raise justifiable arguments.

Response: To address this comment we have included a paragraph in the Introduction Section (line 65-72) that further argues the rationale for the study. Despite there not being any nationally representative studies on the subject, we do comment on the gaps of existing studies the reviewer suggested we mention. We have commented on studies conducted in other contexts – both quantitative and qualitative. This additional paragraph arguing the justification of our study is seen between lines 104-109. 

5. There is no commentary on the findings based on the GMHS 2007 and 2017 at the introduction section. This makes the section scanty on relevant issues.

Response: A commentary on relevant findings based on the 2007 and 2017 GMHS have been included in the introduction section from lines 78-97.

6. In discussing the subject with the socio-ecological model (Lines 132-160), the authors indicated three levels of interests: individual (intrapersonal), partner (interpersonal) and institutional or structural levels. The authors must therefore put the discussion under each of these interests to interpret it within the context of the model.

Response: We appreciate this suggestion. The discussion has been placed under the individual (intrapersonal), partner (interpersonal) and institutional or structural levels in the context of the socio-ecological model at lines 308-379. 

7. At Lines 154-156, the authors indicate that the study is an explanatory study. This is questionable because there are not additional qualitative information to complement the predictor information.

Response: We understand the reviewer’s concern about our description of the study as explanatory. We mentioned this due to the fact that our study’s aim was to examine relationships between the various factors associated with immediate and subsequent contraceptive use among abortion seekers in Ghana. By examining associations between the various independent and dependent variables we apply nomothetic explanation principles. The idiographic explanations are not possible since the data are secondary and quantitative and cannot fully get all the in-depth factors associated with immediate and subsequent use of contraception post-abortion. 

8. Information at Lines 158-160 defeat the earlier position of the author with regard to the justification of the study at Lines 77-88.

Response: We admit that the information provided to justify the study on lines 158-160 contradicts the justification provided on lines 77-88. This error was not intended and has been removed accordingly. 

9. There are no references under the Material and Methods section; not even the source of the data sets. This is unethical or suggests plagiarism.

Response: We regret this oversight and have included a reference of the 2017 Ghana Maternal and Health Survey report on line 182.

10. The socio-ecological model should be discussed with the findings. This was not done at the discussion level.

Response: We are grateful for this observation. The discussion of findings has been sectioned under the individual (intrapersonal), partner (interpersonal), and structural or institutional levels now from lines 308-379.

---

## [Decision Letter · Decision Letter 1]

19 Apr 2021

PONE-D-20-22759R1

Women’s post-abortion contraceptive use: are predictors the same for immediate and future uptake of contraception? Evidence from Ghana.

PLOS ONE

Dear Dr. Kayi,

Thank you for submitting your manuscript to PLOS ONE. After careful consideration, we feel that it has merit but does not fully meet PLOS ONE’s publication criteria as it currently stands. Therefore, we invite you to submit a revised version of the manuscript that addresses the points raised during the review process.

We look forward to receiving your revised manuscript.

Kind regards,

Eugene Kofuor Maafo Darteh, Ph.D.

Academic Editor

PLOS ONE

Reviewers' comments:

Reviewer's Responses to Questions

**Comments to the Author**

1. If the authors have adequately addressed your comments raised in a previous round of review and you feel that this manuscript is now acceptable for publication, you may indicate that here to bypass the “Comments to the Author” section, enter your conflict of interest statement in the “Confidential to Editor” section, and submit your "Accept" recommendation.

Reviewer #1: (No Response)

Reviewer #2: All comments have been addressed

2. Is the manuscript technically sound, and do the data support the conclusions?

Reviewer #1: Partly

Reviewer #2: Yes

3. Has the statistical analysis been performed appropriately and rigorously? 

Reviewer #1: No

Reviewer #2: Yes

4. Have the authors made all data underlying the findings in their manuscript fully available?

Reviewer #1: Yes

Reviewer #2: Yes

5. Is the manuscript presented in an intelligible fashion and written in standard English?

Reviewer #1: Yes

Reviewer #2: Yes

6. Review Comments to the Author

Reviewer #1: Women’s post-abortion contraceptive use: are predictors the same for immediate and future uptake of contraception? Evidence from Ghana.

General Comments

Abortion (Safe and unsafe) generally have arisen from unintended pregnancies. These unintended pregnancies could have been prevented by use of contraceptives. Post –abortion family planning is another window of opportunity to improve CPR. Finding predictors of immediate and later post-abortion contraceptive usage would go a long way to target interventions appropriately.

This study is important for various reasons including the use of socio-ecological model which enables determination of predictors from individual to structural levels. Additionally the use of nationally representative data enables generalization of findings across the country.

The use of secondary DHS data is acceptable. DHS methodologies are robust and standardized across countries over the years to enable comparison of results of similar studies across different countries.

The authors however need to address the following concerns:

1. The assumption that “Given method of contraception at time of abortion as directed by a health provider” is assumed to be:

a. the same as actual contraceptive use and;

b. a measure of immediate post-abortion contraceptive use,

need to be further explained.

2. What post abortion time intervals define immediate and subsequent or later contraceptive use

3. The authors assume current contraceptive use at the time of study to be equivalent to subsequent or later contraceptive use. However, some further clarifications need to be made concerning:

a. How many of the 1750 immediate PAFP users continued or discontinued contraceptive use? Of those who continued use, would the authors still include them as later or subsequent PAFP users?

b. Of those 1750 who discontinued use (If data is available), how many went on to re-use contraceptives subsequently?

c. Did some of the 130 non-users of PAFP (immediate) go on subsequently to use contraceptives or not?

4. In the binary logistic regression analysis, the authors did not specify whether it was a bivariate or multivariate.

5. Further information could have been derived from a multivariate

Reviewer #2: After comparing the revised manuscript with the review comments, It is satisfactory to say that the authors have incorporated all the suggestions and comments into the revision. this makes the article scientifically appropriate for acceptance.

7. PLOS authors have the option to publish the peer review history of their article (what does this mean?). If published, this will include your full peer review and any attached files.

Reviewer #1: **Yes: **Sebastian Eliason

Reviewer #2: **Yes: **Kobina Esia-Donkoh

---

## [Author Response · Author response to Decision Letter 1]

3 Jun 2021

The Academic Editor

PLOS ONE

Dear Dr Darteh,

My co-authors and I are grateful for the opportunity to revise and resubmit our manuscript titled, “Women’s post-abortion contraceptive use: are predictors the same for immediate and future uptake of contraception? Evidence from Ghana” with manuscript number, PONE-D-20-22759R1. 

We are writing to address the comments of the reviewers and provide your office with the revised manuscript. Please see the responses to the comments below.

Best regards,

Dr Esinam Kayi (Corresponding Author)

Reviewer #1: 

1. The assumption that “Given method of contraception at time of abortion as directed by a health provider” is assumed to be:

a. the same as actual contraceptive use

Response: We made this assumption on the basis of the question in the dataset “did they give you the method of contraception, give you a prescription, or give you a referral?” and the corresponding options were ‘gave method’, ‘prescribed a method’, ‘gave a referral’ or ‘don’t know’. We therefore used ‘gave method’ as our measure of immediate post-abortion contraceptive use and assumed it to be actual contraceptive use and acceptance of contraception at the time of post-abortion care. Furthermore, among all the questions on post-abortion contraception, this was the only closely related and direct question asked of the post-abortion women. From these options, ‘gave method’ presupposes that the woman initiated/adopted/accepted a family planning method prior to being discharged after the abortion procedure. Given the data restrictions on the measurement of actual immediate contraceptive use after abortion, we have included this limitation in the manuscript on page 20 and modified the measures section related to immediate post-abortion contraceptive use on page 8. 

b. a measure of immediate post-abortion contraceptive use

Response: In our study, immediate post-abortion contraceptive use was measured by the question “did they give you the method of contraception, give you a prescription, or give you a referral?” (we paraphrased the question as “given method of contraception after abortion”). According to the World Health Organization (2006, 2010), immediate use of post-abortion family planning is measured as a woman’s receipt of family planning immediately after abortion at the health facility before discharge; and or use of contraception within six months post abortion before a subsequent pregnancy. We chose to use WHO’s recommendation of immediate post-abortion contraceptive use as the dataset does not specify the timing of women’s initiation and acceptance of contraceptive use after abortion. Again, the question that most closely measures the immediacy of initiation or use of a contraceptive method is “did they give you the method of contraception, give you a prescription, or give you a referral?” We find that other studies such as Delvaux et al (2008) and McDougall et al (2009) also measure immediate post-abortion contraceptive use based on WHO’s recommendation of post-abortion contraceptive use. 

2. What post-abortion time intervals define immediate and subsequent or later contraceptive use?

Response: The time intervals that define immediate contraceptive use after abortion is initiation or adoption of family planning immediately after abortion at the health facility before discharge; and or use of contraception within six months post abortion before a subsequent pregnancy; and subsequent or later contraceptive use after abortion is contraceptive use 6 months after abortion. For this study, From the 2017 GMHS dataset we used, the century month code for the most recent abortion was not included so we were unable to identify how many months ago women had an abortion prior to their use of contraception at the time of the survey. The information we were able to calculate was limited to the number of years ago the women had the last abortion. We found that 12.6% of women reported having an abortion within a year (or 0 years) preceding the survey. Out of this, 12.6% (or 30 women) were immediate users and 39.2% (or 93 women) were currently using a contraceptive method at the time of the survey (subsequent users). Due to this limitation of not being able to detect the actual period in months, we decided to still include these women as subsequent users of contraception as they reported current use at the time of the survey.

3. The authors assume current contraceptive use at the time of study to be equivalent to subsequent or later contraceptive use. However, some further clarifications need to be made concerning: 

a. How many of the 1750 immediate PAFP users continued or discontinued contraceptive use? Of those who continued use, would the authors still include them as later or subsequent PAFP users? 

Response: We were interested in the women who initiated contraception after abortion for the immediate users. From the analysis, 1750 women did not qualify to be categorized as immediate contraceptive users post abortion as only 130 women reported being given a method of contraception post-abortion. Therefore, if they continued or discontinued contraceptive use, they would be classified as subsequent PAFP users which was considered a separate event. 

b. Of those 1750 who discontinued use (if data is available), how many went on to re-use contraceptives subsequently?

Response: From our categorization of immediate and subsequent post-abortion contraceptive users, 1750 of these women did not qualify to be classified as immediate contraceptive users following abortion. Unfortunately, the data does not allow us to analyze the proportion of women who discontinued contraceptive use in detail. From the dataset, we are only able to identify those reporting use of a method immediately and then subsequently. We found that out of the 1750 women who were not immediate users of contraception, 809 (or 46.2%) were subsequent users of post-abortion contraception. 

c. Did some of the 130 non-users of PAFP (immediate) go on subsequently to use contraceptives or not? 

Response: The 130 women were immediate PAFP users and of this number, 62.9% (82 out of 130) continued contraceptive use (subsequent users) while 37.1% (48 out of 130) were not subsequent users.

4. In the binary logistic regression analysis, the authors did not specify whether it was a bivariate or multivariate.

Response: In the data analysis section of the paper on page 9, we stated that we carried out multivariate analysis using two binary logistic regression models to ascertain the odds adjusting for all the factors which we deemed to be associated with immediate and subsequent post-abortion contraceptive use. However, we have modified and clarified the sentence slightly on page 14 to highlight that the results are from multivariate analyses. 

5. Further information could have been derived from a multivariate

Response: We performed two separate multivariate regression models to examine the predictors (or factors) associated with the two primary outcome variables (immediate and subsequent post abortion contraceptive use). We believe this satisfies the information derived from the multivariate analyses.

Reviewer #2: 

After comparing the revised manuscript with the review comments, it is satisfactory to say that the authors have incorporated all the suggestions and comments into the revision. This makes the article scientifically appropriate for acceptance.

Response: We are grateful to the reviewer for these positive comments.

---

## [Decision Letter · Decision Letter 2]

23 Nov 2021

Women’s post-abortion contraceptive use: are predictors the same for immediate and future uptake of contraception? Evidence from Ghana.

PONE-D-20-22759R2

Dear Dr. Kayi,

We’re pleased to inform you that your manuscript has been judged scientifically suitable for publication and will be formally accepted for publication once it meets all outstanding technical requirements.

Kind regards,

Zelalem Nigussie Azene, MPH

Academic Editor

PLOS ONE

Additional Editor Comments (optional):

Reviewers' comments:

Reviewer's Responses to Questions

**Comments to the Author**

1. If the authors have adequately addressed your comments raised in a previous round of review and you feel that this manuscript is now acceptable for publication, you may indicate that here to bypass the “Comments to the Author” section, enter your conflict of interest statement in the “Confidential to Editor” section, and submit your "Accept" recommendation.

Reviewer #1: All comments have been addressed

Reviewer #2: All comments have been addressed

2. Is the manuscript technically sound, and do the data support the conclusions?

Reviewer #1: Yes

Reviewer #2: Yes

3. Has the statistical analysis been performed appropriately and rigorously? 

Reviewer #1: Yes

Reviewer #2: Yes

4. Have the authors made all data underlying the findings in their manuscript fully available?

Reviewer #1: Yes

Reviewer #2: Yes

5. Is the manuscript presented in an intelligible fashion and written in standard English?

Reviewer #1: Yes

Reviewer #2: Yes

6. Review Comments to the Author

Reviewer #1: In my opinion the authors have responded to the concerns raised. The statistically analysis has been satisfactorily done

Reviewer #2: Article Type: Research Article

Manuscript #: PONE-D-20-22759

Title: Women’s post-abortion contraceptive use: are predictors the same for immediate and future uptake of contraception? Evidence from Ghana.

Authors: Esinam Afi Kayi, PhD; Adriana Andrea Ewurabena Biney; Naa Dodua Dodoo; Charlotte Abra Esime Ofori; Francis Nii-Amoo Dodoo

If the article type is not Research Article, please view this page for more information on other article types: https://journals.plos.org/plosone/s/other-article-types

Abstract

The study recommends an improvement in health system level factors to meet the needs of women at the point of abortion care delivery. However, the results did not point to poor health system factor with respect to the subject.

Socio-ecological model explaining contraceptive behavior among abortion seekers

The content on Lines 130-152 are useful. However, these could have come at the introduction section, immediately after the sentence on Lines 98-99.

Results and discussion

Individual level factors of post-abortion contraceptive use

The discussion was well done. However, it was not discussed with the ecological model.

Structural /institutional level- health system factors

The discussion was well done. However, it was not discussed with the ecological model.

Conclusion

Three issues are found in the Conclusion – conclusion, recommendation and challenges. I think they should stand alone. However, the challenges could come after data analysis.

7. PLOS authors have the option to publish the peer review history of their article (what does this mean?). If published, this will include your full peer review and any attached files.

Reviewer #1: No

Reviewer #2: No

---

## [Editor Report · Acceptance letter]

10 Dec 2021

PONE-D-20-22759R2 

Women’s post-abortion contraceptive use: are predictors the same for immediate and future uptake of contraception? Evidence from Ghana. 

Dear Dr. Kayi:

I'm pleased to inform you that your manuscript has been deemed suitable for publication in PLOS ONE. Congratulations! Your manuscript is now with our production department. 

Kind regards, 

on behalf of

Dr. Zelalem Nigussie Azene 

Academic Editor

PLOS ONE